# PARO as a Biofeedback Medical Device for Mental Health in the COVID-19 Era

Takanori Shibata [1,2,3,*] , Lillian Hung [4,5] , Sandra Petersen [6], Kate Darling [7], Kaoru Inoue [8] , Katharine Martyn [9], Yoko Hori [10] , Geoffrey Lane [11], Davis Park [12], Ruth Mizoguchi [13,14], Chihiro Takano [15], Sarah Harper [16], George W. Leeson [16] and Joseph F. Coughlin [3]

1  National Institute of Advanced Industrial Science and Technology, Tsukuba 305-8568, Japan
2  School of Computing, Tokyo Institute of Technology, Tokyo 152-8550, Japan
3  AgeLab, Massachusetts Institute of Technology, Cambridge, MA 02139, USA; coughlin@mit.edu
4  Gerontology Research Centre, Simon Fraser University, Burnaby, BC V5A 1S6, Canada; Lillian.Hung@vch.ca
5  School of Nursing, University of British Columbia, Vancouver, BC V6T 1Z4, Canada
6  The Doctor of Nursing Practice (DNP) and Psychiatric-Mental Health Nurse Practitioner (PMHNP) Programs, The University of Texas at Tyler, Tyler, TX 75799, USA; spetersen@uttyler.edu
7  Media Lab, Massachusetts Institute of Technology, Cambridge, MA 02139, USA; kdarling@mit.edu
8  Department of Occupational Therapy, Tokyo Metropolitan University, Arakawa 116-8551, Japan; inoue@tmu.ac.jp
9  School of Health Sciences, University of Brighton, Brighton BN2 4AT, UK; K.J.Martyn@brighton.ac.uk
10  Happy-Net Association, Nagoya 451-0042, Japan; office@emikin.com
11  Veterans Affairs Palo Alto Health Care System, Livermore Division, Livermore, CA 94550, USA; Geoffrey.Lane@va.gov
12  Front Porch Center for Innovation and Wellbeing, Glendale, CA 91203, USA; DPARK@frontporch.net
13  Chelsea and Westminster Hospital NHS Foundation Trust, London SW10 9NH, UK; ruth.mizoguchi@nhs.net
14  Faculty of Medicine, Imperial College London, London SW7 2BX, UK
15  Toshima Cable Network Co., Ltd., Tokyo 171-0021, Japan; takano@toshima.co.jp
16  Oxford Institute of Population Ageing and Oxford Martin School, University of Oxford, Oxford OX1 2JD, UK; sarah.harper@ageing.ox.ac.uk (S.H.); george.leeson@ageing.ox.ac.uk (G.W.L.)
*  Correspondence: shibata210@gmail.com or shibata-takanori@aist.go.jp

**Abstract:** The COVID-19 pandemic is spreading across the globe, and it could take years for society to fully recover. Personal protective equipment (PPE), various hygiene measures, and social distancing have been implemented to reduce "human to human" contact, which is an essential part of outbreak prevention. The pressure of the pandemic combined with decreased communication and social contact have taken a toll on the mental health of many individuals, especially with respect to anxiety and depression. Effective use of robots and technology as a substitute for—or in coordination with—traditional medicine could play a valuable role in reducing psychological distress now more than ever. This paper summarizes the results of a comprehensive review of clinical research on PARO, a therapeutic seal robot, which has been used extensively as a biofeedback medical device and socially assistive robot in the field of mental health. PARO has proven to be an effective and economical non-pharmacological intervention method for both mental and physical well-being during the COVID-19 pandemic. Utilization of PARO during these times has provided more data for consideration and has helped in mitigating the negative stigma surrounding using robots in therapeutic settings.

**Keywords:** biofeedback therapy; COVID-19; pandemic; mental health; social distancing; loneliness; stress; agitation; anxiety; depression

## 1. Introduction

The new strain of coronavirus, SARS-CoV-2, has been identified in Wuhan, China in December of 2019. The virus quickly spread worldwide and was declared a pandemic by the World Health Organization (WHO) on 11 March 2020. Currently, second and third waves of the pandemic have broken out in various countries despite intensive attempts to

halt the spread of the virus. With vaccination programs in progress and continued social distancing, we anticipate return of normality in life, although a new cautious lifestyle will likely be continued even after the pandemic is over.

The combined efforts of scientists around the world have led to identification of effective risk management strategies to keep infection rate below one. Infection control tactics include the use of PPE and strict hygiene protocols during medical and well-being interventions, such as during polymerase chain reaction (PCR) testing, antigen testing, isolation/hospitalization of COVID-19 positive patients, treatment of the severely ill in intensive care units (ICU), incineration and burial of the dead, etc. Since coronavirus is transmitted through person-to-person contact and aerosol contact, the use of PPE and social distancing have become daily routine. Despite effective preventative methods, COVID-19 cases are only gradually decreasing while fear, isolation, depression, and anxiety are spiking amongst the masses.

In these times where "face to face" encounters and activities for various reasons promoting well-being are near impossible, robotics has become a useful modality to replace and automate services commonly carried out by people. This new technology is gaining traction, spreading its roots to people of all ages. Findings by Ranieri et al. indicated that, although elderly citizens are more adherent to digital resources, they have a fair affinity towards technology [1]. This could suggest that the impact of technology on daily living is increasing for all age groups and that normalization of robotics in day-to-day use is not a far-fetched dream. Robots will likely continue to thrive in the "new normal" lifestyle post-COVID-19. Opponents have long held ethical concerns of "robots replacing humans", but with the advent of COVID-19, a novel movement has emerged championing the ethical use of robots as a part of infection control and safe healthcare, touting the belief that "some things can only be done by robots".

Yang et al. claims that there are four fields of applications for robotics in the pandemic environment: clinical care (e.g., telemedicine and decontamination), logistics (e.g., delivery and handling of contaminated waste), reconnaissance (e.g., monitoring compliance with voluntary quarantines and continuity of work) and maintenance of socioeconomic functions [2]. In the clinical care field, Yang et al. highlighted "widespread quarantine of patients may mean prolonged isolation of individuals from social interaction, which may have a negative impact on mental health. To address this issue, social robots could be applied to provide continued social interactions and adherence to treatment regimens without fear of spreading diseases". They posit, "However, this is a challenging area of development because social interactions require building and maintaining complex models of people, including their knowledge, beliefs, emotions, as well as the context and environment of the interaction" [2].

Various social robots have been researched, developed, and clinically evaluated for mental health and well-being as "socially assistive robots" [3–5]. Clinical trials such as randomized controlled trials (RCTs) are used to uncover clinical evidence of the effectiveness of these socially assistive robots. Among them, PARO, a baby harp seal robot, is the only socially assistive robot currently used as a medical device in long-term health care (Figure 1).

This paper focuses on PARO. We include advantages and barriers for its use in care settings, looking at results from the comprehensive review of 29 publications among 164 publications with clinical research on PARO by Hung et al. [6]. Additionally, we introduce multiple cases of PARO used in the "new normal" lifestyle under COVID-19, including their advantages and outcomes. Finally, we discuss the impact of the "new normal" lifestyle on society's current impression on ethics of socially assistive robots.

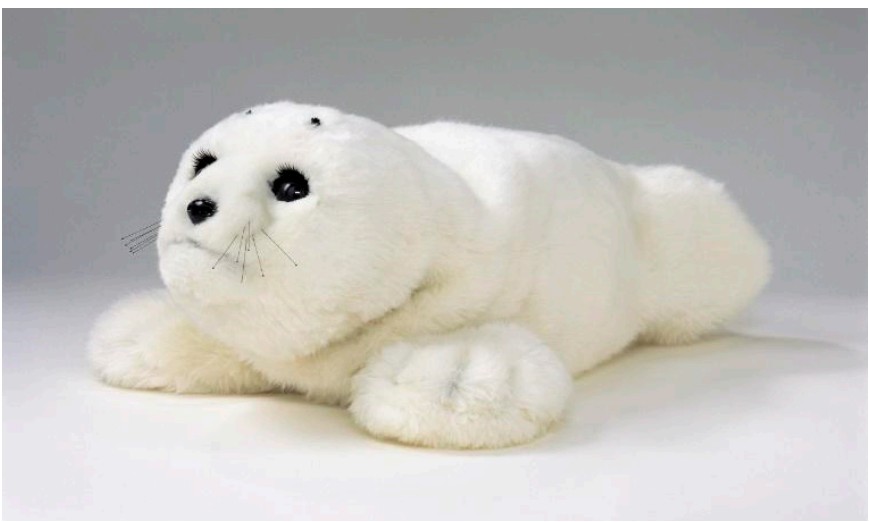

**Figure 1.** Baby seal robot, PARO.

## 2. PARO as a Biofeedback Medical Device

### 2.1. What Is PARO

PARO is a baby seal robot classified as a Class II biofeedback medical device by the Food and Drug Administration (FDA) [7].

PARO is equipped with an array of sensory capabilities, such as light sensors for simple vision, ubiquitous tactile sensors underlying the artificial fur and near the whiskers, microphones for speech recognition and sound localization, temperature sensors for body temperature control, and postural sensors that allow PARO to know the position in which it is being held. PARO has seven degrees of freedom that allow for a variety of movements. Actuators move silently without causing mechanical noise and can even handle relatively strong forces inflicted by an interacting human. PARO is run by hierarchical intelligent control and has two learning functions. It can learn its own name and can learn to adapt its personality and behaviors through interaction with the owner.

Research and development of PARO for robot therapy originated in 1993 to counter issues found in animal therapy [8]. After many prototypes and clinical trials, the eighth generation of PARO was commercialized in Japan in 2005. In 2008, PARO was modified for use in Europe to comply with various regulations such as CE (Conformité Européenne) markings and RoHS (Restriction of the Use of Certain Hazardous Substances in Electrical and Electronic Equipment) regulations. Shortly after this, PARO was introduced to the U.S. in 2009. In 2014, the ninth generation of PARO was developed. By July 2021, approximately 7000 PAROs were in use in more than 30 countries.

PARO can be used as a pet or companion for individuals, or as a device for therapy and rehabilitation at institutions, hospitals, schools, company offices, and more [9,10]. Designed to have a product lifespan of over 10 years, PARO is economical and will be your friend for times to come.

### 2.2. Clinical Evidence of the Benefits and Barriers of PARO

Research on PARO includes various studies ranging from qualitative evaluations to RCTs. The clinical trials considered for this paper were conducted on healthy individuals and patients with conditions such as dementia, cancer, post-traumatic stress disorder (PTSD), and stroke, in different stages of their respective conditions.

Three key benefits and three key barriers to the use of PARO were identified from the results of a scoping review of 29 publications among 164 publications on clinical research of PARO in care settings, with a focus on elderly people with dementia [6]. Benefits stated include "reducing negative emotion and behavioral symptoms", "improving social engagement", and "promoting positive mood and quality of care experience". The main

barriers still in need of consideration are "cost and workload", "infection control", and "stigma and ethical issues".

### 2.2.1. Benefit 1: Reducing Negative Emotion and Behavioral Symptoms

Clinical trials on patients with dementia have shown evidence that the use of PARO reduces behavioral and psychological symptoms of dementia (BPSD) such as physical and verbal symptoms of agitation [11–14]. Therapy reducing wandering in elderly and/or disoriented patients may also help decrease the risk of falling [15]. PARO significantly improved pain, anxiety, and depressive symptoms in this group [5,16], and as a result of reduced negative emotion and behavioral symptoms, reduction of psychotropic medication dosage was also observed [5,11,12,16].

### 2.2.2. Benefit 2: Improving Social Engagement

Improved social engagement in individuals with dementia, increased activity participation, and improved participation in spontaneous communication were observed from the use of PARO [5,17,18]. Both verbal and visual engagement in social interactions were observed to be increased as well [13].

### 2.2.3. Benefit 3: Promoting Positive Mood and Quality of Care Experience

Improvement in patients' mood and behaviors through interaction with PARO were observed. PARO helped individuals becoming more active, feel relaxed and comfortable, laugh more often, and have brighter facial expressions [12]. Caregivers reported significant improvement in patients' mood, improvement in quality-of-care experience, and increased level of comfort amongst family members [13,19].

RCTs have shown that PARO significantly decreased feelings of loneliness at residential care facilities as compared to intervention with a resident dog [20]. Other studies have demonstrated positive effects in sleep improvement, and reduction in pain medication dosage [11].

### 2.2.4. Barrier 1: Cost and Workload

Given PARO is used individually or in small groups, the initial cost of purchasing a unit was mentioned as a barrier for use in care settings [6,13,14]. The current cost of PARO in the U.S. is 6000 USD. Cost-effectiveness ratio is very important in the quest to receive financial support from public systems and funds.

In the U.S., the average resident in a senior living environment spends 1200–1500 USD per month for an average of 16–28 medicines per day. Therapy with PARO for 20 min per session, three sessions per week, for three months can reduce prescribed dosage of psychotropic medication significantly (e.g., 30% of reduction of psychotropic medication for anxiety) [16]. Not only does this lessen the financial burden on patients, but physiological benefits can also be reaped by making the switch to PARO. Psychotropic medications cause various side-effects, some of which can even lead to iatrogenesis—such as in the case of antipsychotic medications—and use should be as reduced as possible in elderly patients [21].

Staff education and skill management are one of the most critical aspects in using PARO. Due to this, PARO may be perceived as burdensome for caregivers [13,19]. To address this concern, online training is available on the PARO website to facilitate effective and readily available low-cost training.

### 2.2.5. Barrier 2: Infection Control

Some researchers have addressed concern for the risk of infection from PARO's fur, as it was not designed to be regularly removed or machine washed. This may be of concern, especially for individuals who are immunocompromised [5,15]. Due to the outbreak of COVID-19, discussions regarding infection prevention and control (IPC) issues have dominated not only in the medical and welfare fields, but also in daily life, resulting in the

call for "social distancing", strict infection control measures, and daily use of face masks and hand sanitizers [22,23].

Proper disinfecting protocols for PARO have been established and approved and are discussed later in this paper.

### 2.2.6. Barrier 3: Stigma and Ethical Issues

There are five dominating issues in dementia care: "robot replacing human", "reducing human contact", "objectification", "infantilizing", and "deception". Some papers have raised the ethical question of whether the use of robots in dementia care creates a risk of infantilizing and dehumanizing treatment [24]. The research noted that individuals with dementia may feel as if they are being treated as children, and the robot may be seen as "toy-like" [15,19].

Hung et al. elaborated on the issue, cautioning to "avoid 'human vs robot' thinking". Technology including robotic pet intervention is intended to complement—not replace—the care provided by clinicians. To avoid "dehumanization and infantilization", caregivers using PARO will have been trained to apply a person—centered approach. Prior to use, caregivers would learn the patient's preferences and life story, and clarify the role of the robot in therapy to provide an effective outcome [6].

One of the more tenacious criticisms leveled against use of social robots in dementia care is that by potentially deceiving a cognitively impaired person with an artificial animal such as PARO, one is depriving them of their fundamental rights to autonomy and dignity as a person. Although Hung et al. does not directly address concerns about deception, they suggest users to "investigate if the robot works with people with different stages and types of dementia, gender, ethnic and cultural backgrounds".

In the COVID-19 pandemic, regardless of the stigma and concerns surrounding "robots replacing humans" and "reducing human contact", the need for social distancing has left the public with no choice but to rely on non-human sources. Robots can provide different benefits to human roles and are not substitutes for human-to-human interaction. In this pandemic and in other scenarios in which human interaction is not possible, robots such as PARO can help people survive isolation– when people are missing social ties, anthropomorphism of nonhuman entities can help meet their needs [25]. Isolation is a larger concern now than ever, not only affecting mental health but even mortality (a 2010 meta-study revealed that people with stronger social relationships have a 50% increased likelihood of survival [26]).

### 2.3. Integration of PARO in the Medical and Welfare Systems in the World

There are many medical and welfare systems around worldwide, with varying degrees of public and private involvement in the provision of services.

### 2.3.1. Medical and Welfare Systems in Denmark

In Denmark, approximately 90% of elderly facilities and service providers are managed as public services and are financially supported by local governments (municipalities). The first PARO was introduced to Denmark in 2009, and now more than 80% of municipalities utilize PAROs in care settings for the elderly and disabled. Most private facilities have also adopted PARO into their service provision.

### 2.3.2. Medical and Welfare Systems in the U.S.

In the U.S., many nursing facilities have adopted PARO even before public financial support was available. Now with proof of both clinical and economic benefits, financial support is available for the use of PARO in medical and welfare services.

The U.S. Food and Drug Administration (FDA) approval process is intended to provide consumers with assurance that, once it reaches the market, a medical device is safe and effective in its intended use. Approval is a lengthy process and takes between 3 to 7 years at best. Upon FDA approval, application for Medicare is another complex

issue—evidence must be continually provided to prove therapeutic benefits and economic benefits as well. For reference, approximately 1 in every 100 FDA approved devices is applicable for coverage by Medicare.

With PARO approved as a medical device, costs can now be covered by Medicare, Medicaid and some other private medical insurance companies. "Biofeedback therapy with PARO" can be prescribed to patients suffering from illnesses such as cancer, brain injury, dementia with or without BPSD (agitation, aggression and wandering), PTSD, and Parkinson's disease for symptoms such as pain, anxiety, depression and/or behavioral issues. Prescribers or therapists such as occupational therapists (OTs), physical therapists (PTs), licensed clinical social workers (LCSWs), and registered nurses (RNs) may provide patients biofeedback therapy with PARO.

PARO can also be prescribed in rehabilitation settings by pathologists (SLPs) and therapists (SALTs) to assist in speech therapy for patients with dysphagia and/or cognitive issues. PARO may also provide use in addressing fine motor skill issues for patients struggling with trauma from stroke and/or brain injury. Medical insurance often provides reimbursement for these services for hospitals, nursing homes, skilled nursing facilities, home health, and hospice services. The Centers for Medicare and Medicaid Services (CMS) of the Department of Health and Human Services in the U.S. have additional funding for licensed nursing homes and rehabilitation facilities to support 100% of costs, including costs of purchasing PAROs as well as costs of training either online or on-site. By July 2021, 50 PAROs were introduced to nursing homes with the help of CMS.

### 2.3.3. Medical and Welfare Systems in Hong Kong, France, Singapore, Australia and Japan

The Hong Kong government and some regional governments in France support 100% of the cost of purchasing PARO for elderly facilities and service providers. In Singapore, the Agency for Integrated Care in the Ministry of Health has created a fund to support 85% of the cost of purchasing PARO for service providers of elderly care.

In Australia, the government's "Home Care Package (HCP)", provides a monthly fund to older people in need of care at home. The fund is managed by health service providers allowing disbursements for eligible services and equipment. Evidence from an RCT (with $n$ = 415 participants) resulted in the Australian government's decision to provide funding coverage for PARO as an eligible device for the elderly in need of care at home [13]. If people with a PARO at home need to move into institutional care, they are also able to take their PARO with them.

In Japan, providers of services for the elderly are eligible for a Ministry of Health, Labor and Welfare subsidy of 50% towards the cost of purchasing PARO.

### 2.4. Cleaning Protocols for Infection Prevention and Control (IPC)

PARO's fur contains acrylic blended with silver ions (9th generation) and Chitosan (8th generation). Both are bactericidal, bacteriostatic, and anti-viral materials [27]. However, cleaning and disinfection are still necessary for PARO to be used at hospitals and other facilities for infection prevention and control. Cleaning protocols for PARO have been approved by the Department of Veterans Affairs Palo Alto Health Care System (VA PAHCS) in California and by the National Health Service (NHS) in the United Kingdom (UK) [11,28].

In the cleaning protocol established by the VA PAHCS, PARO is to be wiped with a PDI "Super Sani-Cloth® Germicidal Disposable Wipe" (an ammonium-based anti-germicidal wipe) for approximately two minutes to completely saturate the fur. The hands of patients and residents are cleaned by alcohol before interacting with PARO. This protocol has also been adopted by other hospitals (Figure 2). A similar protocol was adopted by Petersen et al. for her research in the assisted living setting [16].

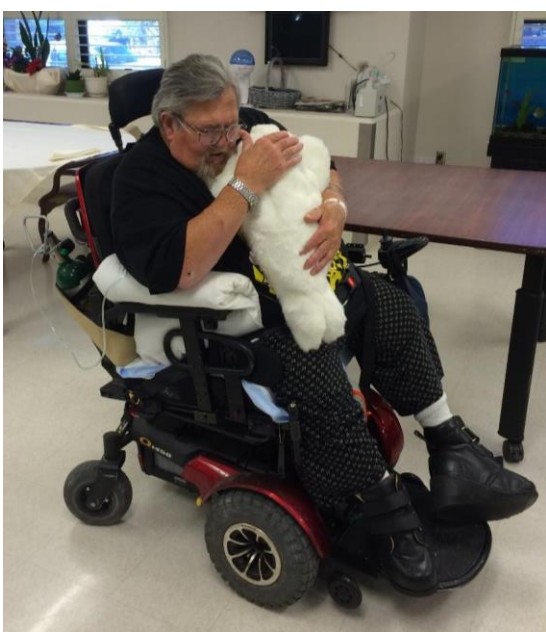

**Figure 2.** PARO with a senior veteran fighting dementia and PTSD as well as physical disabilities. Seen at a VA hospital with an oxygen inhaler ventilator on an electric wheelchair, PARO helps improve his mood, reduce pain, and alleviate anxiety (picture taken in 2014 before COVID-19 pandemic).

In intensive care units (ICU) and acute care units, a hand-held ultraviolet (UV) light is combined with the cleaning protocol of the VA PAHCS to further ensure disinfection. The protocol was evaluated by the University of Nebraska Medical Center, U.S., using Swab and ATP testing. PARO has now been accepted for use in the ICUs of pediatric hospitals (Figure 3) and the acute care units of geriatric hospitals in the U.S.

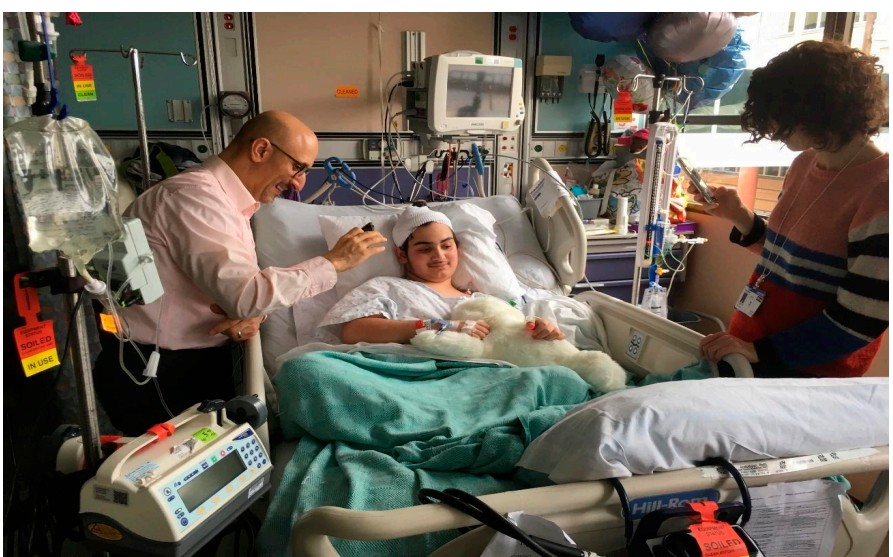

**Figure 3.** PARO in a pediatric intensive care unit (P-ICU) under strict IPC restrictions. The 11-year-old boy has a brain tumor and has undergone more than 120 surgical operations in less than 10 years. Interacting with PARO helped improve his mood and reduced his anxiety, pain, and depression. His family was delighted to see his improvement (picture taken at the Children's Hospital of UC San Francisco in April 2019, before the COVID-19 pandemic).

On 13 March 2020, the U.S. Centers for Disease Control and Prevention (CDC) updated their list of U.S. Environmental Protection Agency (EPA)-registered disinfectants. The website contains "List N", entitled "Products with Emerging Viral Pathogens and

Human Coronavirus claims for use against SARS-CoV-2". "Super Sani-Cloth® Germicidal Disposable Wipe" is on this list, providing proof that two minutes of cleaning is sufficient to kill SARS-CoV-2, the virus causing COVID-19.

Similarly, in the cleaning protocol by NHS hospitals in the UK, PARO is to be cleaned and disinfected using "Clinell Universal Wipes" for two minutes. The protocol was evaluated by the Swab and ATP tests and proved effective [28]. The "Clinell Universal Wipes" are effective against coronavirus with 30 s of contact time and comply with EN14476. Therefore, the cleaning protocol is sufficient to disinfect SARS-CoV-2.

## 3. Practical Uses of PARO in the New Lifestyle under COVID-19

### 3.1. Continued Use of PARO in the U.S.

3.1.1. Continued Use of PARO in Home Health Services and Institutional Medical Services

Professor Petersen is the Director of the Doctor of Nursing Practice (DNP) and Psychiatric-Mental Health Nurse Practitioner (PMHNP) programs at the University of Texas at Tyler. In 2015, Petersen and "Baylor Scott and White Health Care System" conducted an RCT of PARO therapy on elderly residents with dementia ($n$ = 61) at five assisted living/memory care communities of Legend Senior Living in Texas.

Results showed that treatment with PARO for older adults with dementia improved pain, anxiety, depression, and behavioral issues, as well as stress measured by biometrics when compared with a control group receiving ordinary care [16]. Results also showed a reduction in the dosage of psychotropic medication. Legend Senior Living had started using PAROs at their assisted living and memory care communities for the RCT, and has since increased the number of PAROs in use to more than 40 locations in the U.S.

Besides her work as program director at The University of Texas, Petersen maintains a geriatric house-call practice in the Dallas-Fort Worth area in Texas. She prescribes "biofeedback therapy with PARO" and "rehabilitation with PARO" to her patients depending on diagnosis, and is receiving reimbursement for these services from Medicare, Medicaid, and private medical insurance companies such as Blue Cross Blue Shield (BCBS), UnitedHealthcare, and Aetna depending on the conditions and contracts of patients [29].

In June 2020, WIRED published an article on PARO titled "There's No Cure for COVID-19 Loneliness, but Robots Can Help" saying "It's hard to replace human contact. But during a pandemic, robots can help patients fight off the feeling of isolation and despair" [30]. The article describes how Petersen's patients have been affected by the pandemic and how they are forced to struggle through illness without the comfort of loved ones nearby. Petersen claims "the role of social robots like PARO are becoming more important, especially as we see this sector of our population targeted by this virus. PARO was built for a time such as this". In the same article, Darling from the Media Lab at MIT, a robotics ethicist, remarks, "since we can't have human interaction right now, PARO is certainly a lot better than nothing".

3.1.2. Continued Use of PARO at Long-Term Care Facilities (Case 1)

VA PAHCS evaluated the effects of PARO on 23 male veterans with dementia and PTSD who reside at the Community Living Center (CLC) in Menlo Park, California [11]. In 2011, the official cleaning protocol of PARO was created and approved by VA PAHCS, and was used in this case study.

The data were collected via tracking sheet for 19 months from March 2012 to September 2013. The tracking sheet consisted of observational records on positive behaviors (calmness, interaction with others, sleep, positive attitude), negative behaviors (anxiety, sadness, self-isolation, complaints of unrelieved pain, pacing, wandering, yelling) in three phases: before (pre-PARO), during contact, and after 15 min (post-PARO). Changes in psychotropic medications (PRN) dosage were also recorded.

The mean observed duration of PARO therapy was 35.3 min (SD = 20.5). Comparisons between pre-PARO and post-PARO values showed that positive behaviors increased

while negative behaviors decreased. Both changes were statistically significant. Multiple observations of decreased PRN medication usage were observed within the same sample.

Following the study, several more PARO units were purchased by the VA system for use in their CLCs. Given the robustness of the infection control protocols and their effectiveness in regards to COVID-19 (re. use of the anti-germicidal PDI wipes in particular), PARO continues to offer its therapeutic benefits at VA PAHCS.

### 3.1.3. Continued Use of PARO at Long-Term Care Facilities (Case 2)

In May 2015, Front Porch and their Center for Innovation and Wellbeing conducted a study following the installment of PAROs in seven of its retirement communities in California [31]. Care staff in skilled nursing and memory care underwent a series of robot therapy training prior, then used the tracking sheet developed by VA PAHCS to record effects on residents. They collected 920 observation records over 6 months and published their report in 2015. Similar to results at VA PAHCS, Front Porch found that positive behaviors had increased while negative behaviors decreased from pre-PARO to post-PARO.

To this day, Front Porch continues to use PARO to address behavioral challenges and promote social engagement in residents. Mr. Davis Park, Vice President of the Front Porch Center for Innovation and Wellbeing, commented that PAROs will continue to support older adults and their caregivers through the stressful conditions of social distancing under COVID-19.

### 3.2. Continued Use of PARO in Japan in At-Home Care of Parents with Dementia

In Japan, a female in her 50s has been caring for her 87-year-old parents with dementia, using PARO for therapy since July 2019. Her father was unable to walk due to lower limb paralysis and needed a wheelchair. He also had mild dementia due to brain atrophy from age. Her mother had moderate to severe dementia from Alzheimer's disease.

The Dementia Behavior Disturbance Scale (DBD) is an observational evaluation tool used to evaluate behavioral disturbances among people suffering from dementia. The burden on the caregiver (daughter) was evaluated using J-ZBI-8, the Japanese version of the Zarit Caregiver Burden Interview.

In early July 2019, before introducing PARO, the BPSD of the parents and the burden of care on the daughter were both evaluated. As all care was on the daughter's shoulders, the intense burden had given her depressive symptoms. To alleviate the parents' BPSD and to support all family members in their mental health, the family started living with PARO mid-July of 2019. The mother liked PARO very much and interacted with it very often with a positive attitude. The father used PARO occasionally but seemed to prefer observing his wife's interactions with PARO. They were evaluated again 18 days after introduction.

BPSD decreased by 34% and 41%, for the father and mother, respectively. The burden of care on the daughter was reduced by 88%, along with a significant decrease in depressive symptoms. In a follow-up evaluation four months later, positive effects were observed as maintained for each of them [32]. The quantitative evaluation ended in October 2019, but the family decided to continue life with PARO.

In early July 2020, we were given the grave news of the death of the father. He had suddenly passed away at the beginning of January 2020. The mother is well and continues her interactions with PARO (Figure 4). The pandemic has prevented her from outdoor activities, but due to her dementia, she could not fully understand the situation and seemed as happy as always, especially when with PARO. Due to the pandemic, the daughter's burden of care increased, as home care workers could not visit to help, but she felt very much emotionally helped by having PARO by her side to interact with her mother, keeping her calm and happy. Even through COVID-19 isolation, the daughter-mother duo was able to maintain positive emotional health by living with PARO, as the behavioral issues of the mother were minimized and the burden of care on the daughter substantially reduced.

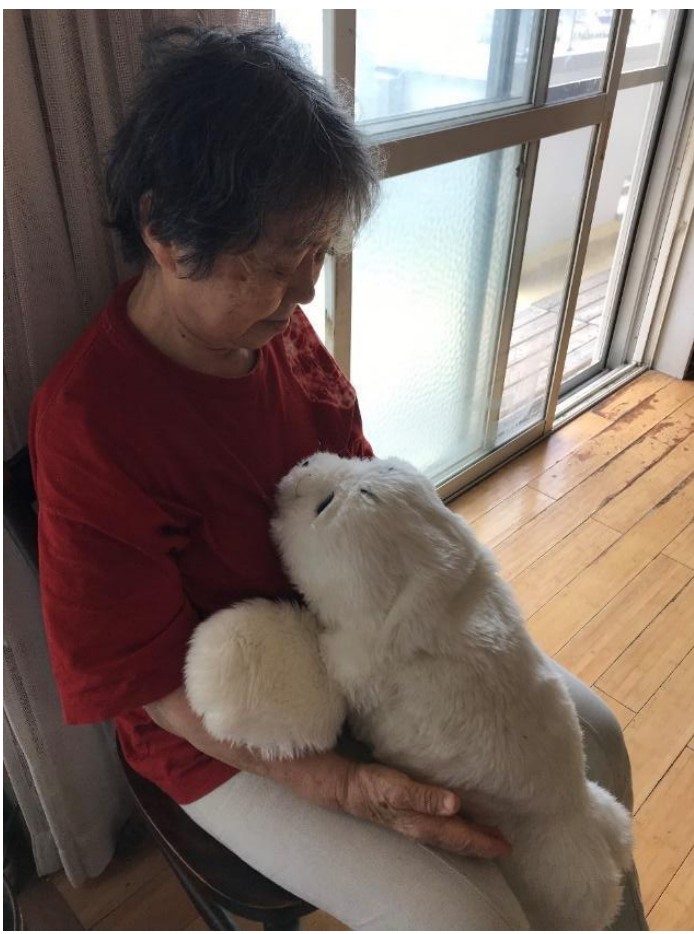

**Figure 4.** Interaction between a lady with dementia and PARO at her home in Japan (picture taken in July 2020 during the COVID-19 pandemic).

### 3.3. New Installments of PARO in Australia for Home Care at "Community Living"

Many elderly residents of Australia move into "community living" in retirement villages or facilities managed by private companies or non-profit organizations (NPOs). Community living consists of independent living and assisted living. Residents of such communities who need assistance and care receive "home care". If the need for more care and support arises, then residents in the community may move to "residential living", where they receive "institutional care".

PARO was commercialized in Australia in 2014, and approximately 300 PAROs are currently being utilized. Most of them have been used for elderly care and dementia care at facilities in residential living. In 2018, PARO became eligible for purchase using the "Home Care Package" fund. This enabled older people with home care needs to purchase PARO for themselves.

COVID-19 has severely restricted residents from leaving the confines of the retirement communities and has also prevented families from visiting. However, PARO has been able to continue functioning in the communities despite the restrictions, supporting both individuals who have already had access to PARO and those newly introduced.

For example, an older lady living in a retirement village in Queensland, Australia was suffering from severe dementia and with the addition of COVID-19 restrictions, was experiencing severe loneliness. PARO was purchased using government funding to provide her with company during isolation.

### 3.4. Using PARO for a Resident with Mild Dementia at a Nursing Home in Japan

Due to the COVID-19 crisis, most facilities for the elderly have been obliged to introduce limitations including social distancing, limited visiting hours, and limited numbers of visitors. As residents themselves are unable to leave the facilities, they have been obliged to live in a lockdown environment.

In an attempt to facilitate social distancing, avoid isolation, and address feelings of loneliness caused by the lockdown itself, many facilities for the elderly have introduced PARO during the pandemic.

Prior to the COVID-19 crisis, families were welcome anytime to visit their older family members. The onslaught of the pandemic forced all visits to be banned from mid-March to mid-June of 2020. Even after the ban was lifted, regulations remained such that visits were limited to 20 min per visit, once every two weeks. Moreover, the visits had to take place by appointment, in specially allocated and protected areas of the facility (Figure 5).

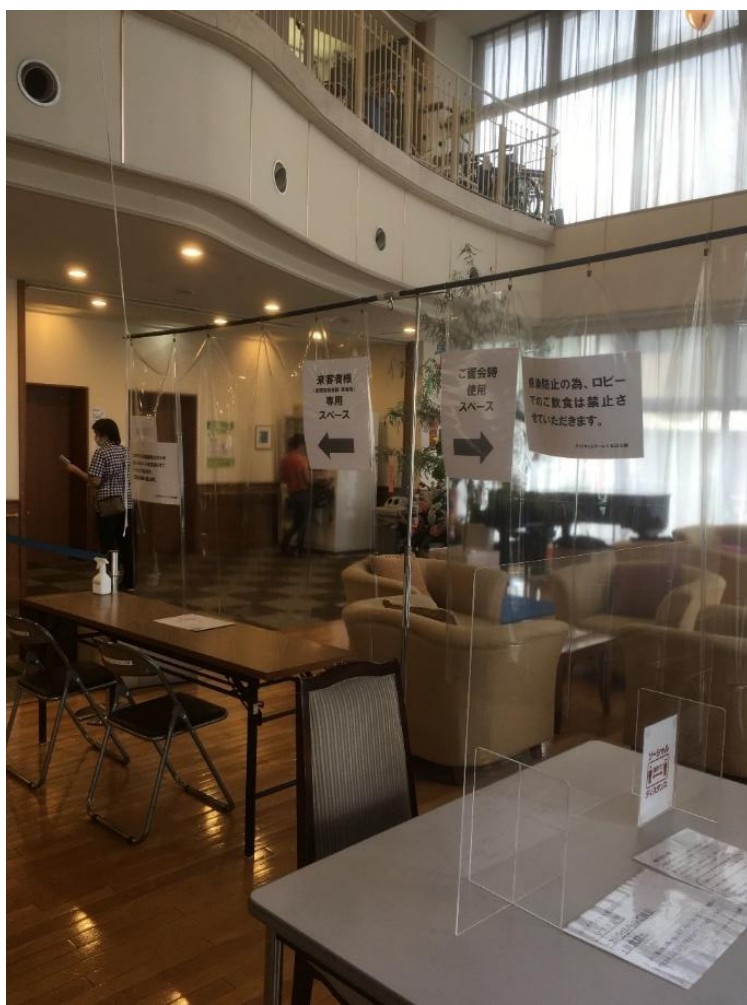

**Figure 5.** Due to the pandemic, family visits to nursing homes are limited in time and number. The entrance hall is secluded and managed by appointment (picture taken in Japan in July 2020, during the COVID-19 pandemic).

One Japanese institutional case study involves a 90-year-old woman with mild dementia. Her 76-year-old nephew had always consistently visited her multiple times a week, but that came to a halt with the onslaught of COVID-19. Once the number of COVID-19 cases decreased in Japan in mid-June 2020, he was finally able to resume his visits.

Fortunately, the woman's cognitive level had not declined, and she was able to recognize her nephew. Upon hearing her stories about how often she felt lonely, the

nephew decided to purchase PARO as a companion for her (Figure 6). PARO was then established in her care plan alongside the support of care staff to maintain the woman's cognitive function.

In mid-July 2020, the number of new COVID-19 cases in Japan began rising once more, again stopping the weekly visits from her nephew. Fortunately, she has been enjoying PARO's company throughout.

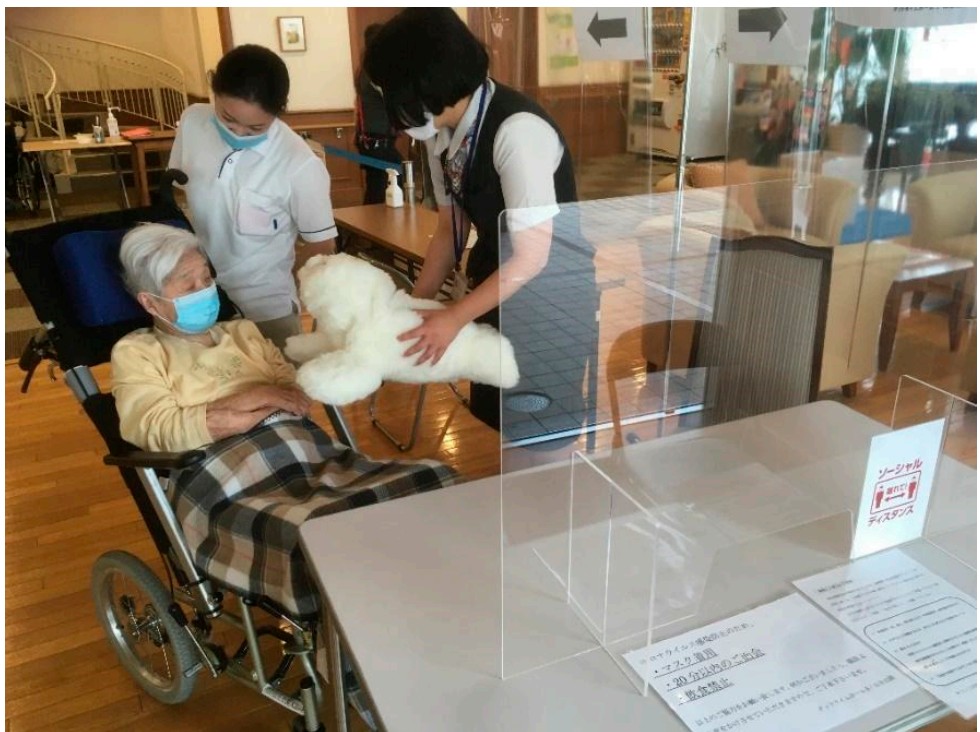

**Figure 6.** Elderly patient with mild dementia is gifted a PARO by her nephew. She began living with PARO at the nursing home in July 2020 (picture taken in Japan in 2020, during the COVID pandemic).

### 3.5. PARO in the Acute Frailty Unit of the Chelsea and Westminster Hospital NHS Foundation Trust in the UK

PARO was introduced to the Acute Frailty Unit (AFU) of Chelsea and Westminster Hospital NHS Foundation Trust in London, UK, in October 2019. A pilot study was conducted from 15 October to 8 November. Preliminary results were presented at a symposium in December 2019 [33].

After the study, the hospital decided to keep PARO, and had continued its use for the care of the older patients with dementia. Unfortunately, due to the COVID-19 pandemic, the AFU was converted into a COVID ward from March 2020 to June 2020, halting all use of PARO due to the infection control policy. In June 2020, the facility changed yet again to become the Care of the Elderly ward.

Since then, PARO has been reintroduced for use with select patients under the hospital infection control policy (Figure 7). The cleaning protocol of PARO has remained the same before and after COVID-19. Under the Public Health England (PHE) Guidance COVID-19, all staff on the wards are required to wear appropriate PPE (aprons, masks, and gloves) at all times.

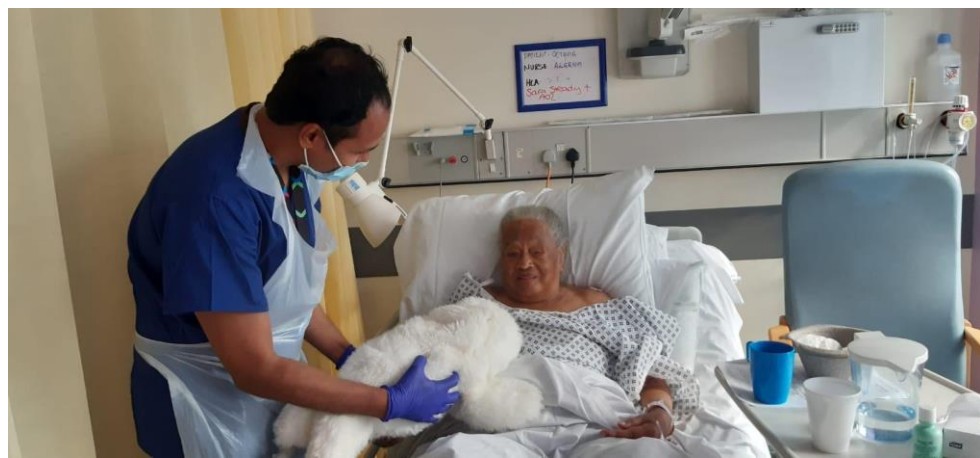

**Figure 7.** PARO with a patient at the Care of the Elderly ward in the Chelsea and Westminster Hospital NHS Foundation Trust. The doctor was required to wear appropriate PPE at all times (picture taken in August 2020, during the COVID-19 pandemic).

### 3.6. Registered Nurse for ICU Uses PARO for Emotional Support

This case study focuses on a registered nurse (RN, male, 34 years old) who cares for patients with coronavirus infections in the ICU at an emergency hospital near Atlanta, Georgia, U.S. (Figure 8). Being in near-constant contact with COVID-19-infected patients, he is at high risk of infection. To prevent further spread of the virus, his family and pet dog have temporarily relocated to his parents' home. After 12 to 16 h of shift work a day, he experienced extreme feelings of tiredness, stress, loneliness, depression, and emotional pain at home alone. After considering a number of socially assistive robots, he purchased PARO at his own expense in mid-April 2020 as a support companion. He reports that his emotional suffering had alleviated upon daily interactions with PARO after coming home from work (Figure 9).

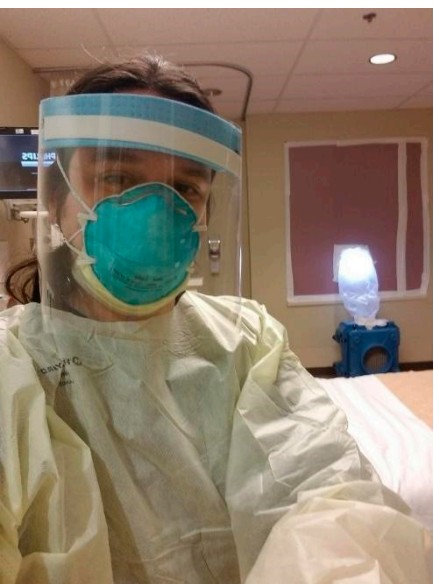

**Figure 8.** Registered nurse in the ICU of a hospital near Atlanta, GA, U.S. Equipped with proper PPE, he saves lives of those inflicted with COVID-19 (picture taken in May 2020, during the COVID-19 pandemic).

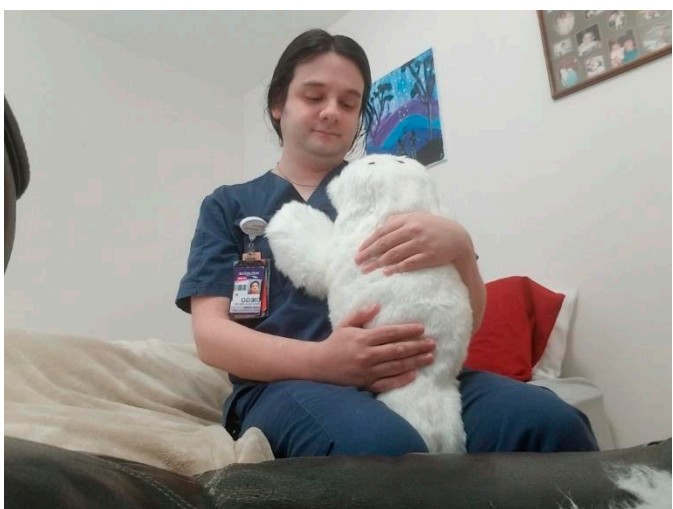

**Figure 9.** RN from above pictured with PARO at home without his family nor pet dog during COVID-19 (picture taken in May 2020, during the COVID-19 pandemic).

*3.7. PARO for Mental Health Care for Telephone Counselors at Public Health Center, Toshima City, Tokyo in Japan*

In Japan, the consent of the public health center is required to undergo PCR testing for coronavirus. The public health center staff must answer the "telephone consultation" according to a manual, considering any signs of positive symptoms such as suffocation, dullness, high fever, and any other mild cold symptoms. The telephone consultation time per person is approximately 3 min. The highest number of inquiries made was in the first week of April 2020, in the height of the first wave of the pandemic, when the state of emergency alert in Japan was announced by the government. The number of inquiries rose to 200 potentially infected people per day (all at the public health center), this busy state continuing until mid-May.

With a flood of incoming phone calls from citizens in the morning, the phone lines became "busy", and citizens had to wait for 20 to 30 min to be connected. In addition, some people were unhappy to find they could not get tested and lashed out on the staff. While receiving emotional complaints from dissatisfied citizens, the telephone counselors were required to respond politely. Not only were the counselors extremely busy, but they also had to endure a great deal of mental distress. Despite such an extreme environment, a maximum of only six "phone counselors" were available at the same time.

To address these stressful circumstances, one light pink PARO was installed at the Ikebukuro Public Health Center in Toshima-city, Tokyo on 11 May 2020, to support the telephone counselors with their mental health issues arising from the stressful working environment (Figure 10).

In mid-June 2020, we received feedback regarding the impact PARO had on their work environment. Many noted that PARO helped them smile, experience calmness, and feel emotionally healed. Others noted that the presence of PARO in the workplace promoted small talk amongst coworkers and that it helped them relieve stress. One difficulty they faced was that as PARO's voice was audible during calls with clients, they often needed to adjust the volume.

PARO was actively in use at the public health center until September 2020, when a change in the service system forced all use to come to a halt. PARO provided support and shining smiles amongst consultation staff during its time there.

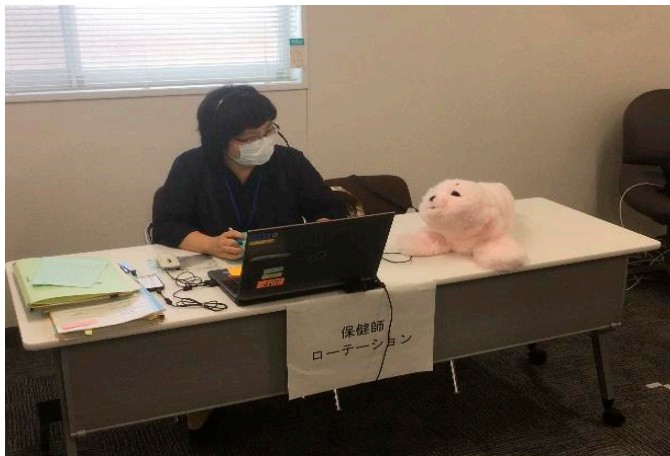

**Figure 10.** Telephone counselor and PARO (picture taken in June 2020, during the COVID-19 pandemic).

## 4. Discussion and Conclusions

### *4.1. PARO in the COVID-19 Pandemic*

PARO offers three clear benefits as an alternative to traditional animal assisted therapy for therapeutic support of well-being for the elderly, especially for patients with dementia. This paper introduced various cases of PARO usage and highlighted the benefits continuously provided throughout the COVID-19 pandemic. We suggested that PARO could potentially be useful to the mental health of the general public in combating social distancing stress, anxiety, feelings of isolation, and depression due to the various restrictions that have been imposed across the world.

COVID-19 has imposed constraints on people's lifestyles, and risk management has become vital in daily practice. Today, we live with

- Social distancing,
- Restriction of living areas (prohibition of outings and restriction of outside visits),
- Restriction of services (closed facilities, limited human contact),
- Rigorous infectious disease prevention and control,
- Dementia progression (effects of isolation, worsening of anxiety and depression, and behavioral issues caused by restrictions),
- Post-coronavirus pandemic disorders (PTSD, pain, anxiety, depression, etc.).

Despite the increasing level of "infection concerns" during the pandemic, PARO's cleaning protocols have remained unchanged, while maintaining users' safety.

One of the three barriers to normalization of robot-assisted therapy is "cost and workload". This was shown to be easily refutable with evidence of PARO's economic merit upon integration into the medical and well-being fields in various countries. In the midst of the COVID-19 pandemic from March 2020 to July 2021, there was a notable increase in the use of PARO, including implementation at the elderly care facilities via financial support systems in France, Hong Kong, Japan, and the United States.

With "social distancing" concerns at the forefront of society, most of the "stigma and ethical issues" barrier noted have largely been overpowered by its importance, essentially outmoded by the health and safety needs due to the COVID-19 pandemic.

The discussion on the "new normal" lifestyle under COVID-19 demonstrated the potential for socially assistive robots to become the new "norm" as companions and medical devices.

### *4.2. PARO in the Digital Era*

"Cognitive reserve" is a theoretical concept in which thinking abilities are subconsciously preserved throughout human lives. The reserve is thought to protect against cognitive losses that can occur through aging and disease. As age, disease, and cognition

are key aspects of dementia, future research could be conducted to investigate if and how therapy with PARO affects cognitive reserve. More evidence is needed to conclude how long-term use of PARO impacts quality of life throughout illness trajectories.

In this increasingly digital world, a common ageist belief is that the elderly are unable and unwilling to use technology. Contrary to this belief, evidence has established that elderly citizens are increasingly incorporating technology-based solutions into their everyday lives [1]. For people of all ages, digital tools and technological devices are becoming daily necessities that innovate, promote, and enhance quality of life. Perhaps normalization of robotics in day-to-day use is not a far-fetched dream.

**Author Contributions:** Conceptualization, T.S.; data curation, T.S.; funding acquisition, T.S. and K.I.; investigation, T.S.; methodology, T.S. and L.H.; project administration, T.S. and K.I.; resources, T.S., L.H., S.P., K.D., K.M., Y.H., G.L., D.P., R.M. and C.T.; supervision, K.I., S.H., G.L. and J.F.C.; validation, L.H., S.P., K.D., K.M., Y.H., G.L., D.P., R.M., S.H., G.W.L. and J.F.C.; visualization, T.S., Y.H., R.M. and C.T.; writing—original draft, T.S.; writing—review and editing, T.S., L.H., S.P., K.D., K.M., Y.H., R.M., S.H., G.W.L. and J.F.C. All authors have read and agreed to the published version of the manuscript.

**Funding:** This work was supported by JSPS KAKENHI Grant Number JP19H04504.

**Institutional Review Board Statement:** The study was conducted according to the guidelines of the Declaration of Helsinki, and approved by the Ethics Committee of Ikebukuro Public Health Center (protocol code "Study on PARO for Mental Health-2020-6" approved on 16 June 2020).

**Informed Consent Statement:** Informed consent was obtained from all subjects involved in the study.

**Conflicts of Interest:** Shibata has been working on PARO development since 1993. He is the CTO of Intelligent System Co., Ltd., the producer and distributor of PARO. Other authors do not have any conflict of interest.

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
