# Peer review of "PARO as a Biofeedback Medical Device for Mental Health in the COVID-19 Era"

_sustainability, doi:10.3390/su132011502_

Round 1

Reviewer 1 Report

I was very happy to review this paper. It is interesting and could be suitable for publishing after some improvements. 

I suggest to improve the introduction and discussion paragraphs by the digital era as prompting for robotic solution using. 

I suggest including in the references and introduction some highlights of following paper:

 - Ranieri, J., Guerra, F., Angione, A. L., Di Giacomo, D., & Passafiume, D. (2021). Cognitive Reserve and Digital Confidence among Older Adults as New Paradigm for Resilient Aging. Gerontology and Geriatric Medicine. https://doi.org/10.1177/2333721421993747

 Even discussion should be improved by digital era implications. 

Reviewer 2 Report

The presented article is an interesting update on the use of the PARO robot as psychological support for people with various health problems. The authors reviewed the existing research on this topic, preparing, on the basis of the analysis, a summary of the advantages and problems related to the use of PARO. The update of information on the use of PARO concerns its application in the conditions of the COVID-19 pandemic. The pandemic turned out to be a period that strongly influenced the psychological state of people, including people struggling with various diseases or difficult situations. The article provides valuable information on the possibilities of psychological support with the use of robots in times of social isolation.

The discussion of the article is quite sparing, the authors could extend it, inter alia, summarizing the conclusions that result from the materials presented.

The article is written in a clear, accessible language, and the presented content is cognitively interesting.
